# Chicken Skin Decontamination of Thermotolerant *Campylobacter* spp. and Hygiene Indicator *Escherichia coli* Assessed by Viability Real-Time PCR

**DOI:** 10.3390/pathogens11060706

**Published:** 2022-06-18

**Authors:** Imke F. Wulsten, Maja Thieck, André Göhler, Elisabeth Schuh, Kerstin Stingl

**Affiliations:** 1National Reference Laboratory for *Campylobacter*, Department of Biological Safety, German Federal Institute for Risk Assessment (BfR), 12277 Berlin, Germany; imke.wulsten@bfr.bund.de (I.F.W.); maja.thieck@bfr.bund.de (M.T.); 2National Reference Laboratory for *Escherichia coli* including Verotoxin-Producing *E. coli*, Department of Biological Safety, German Federal Institute for Risk Assessment (BfR), 12277 Berlin, Germany; andre.goehler@bfr.bund.de (A.G.); elisabeth.schuh@bfr.bund.de (E.S.)

**Keywords:** VBNC, propidium monoazide, qPCR, food safety, *uidA*, β-glucuronidase, peroxyacetic acid, lactic acid, freezing, internal sample process control, IPIU

## Abstract

Thermotolerant *Campylobacter* spp. are fecal contaminants of chicken meat with serious implications for human health. *E. coli* is considered as hygiene indicator since, in contrast to *Campylobacter*. spp., the bacterium is generally present in the avian gut. Stress exposure may transiently cease bacterial division. Therefore, colony forming units (CFU) may underestimate the infection risk of pathogens. We developed a viability real-time PCR (v-qPCR) for the quantification of viable *E. coli* targeting the *uidA* gene, encoding β-glucuronidase, which is usually detected for phenotypic species identification. The short- and long-term effects of decontaminating chicken skin on the survival of both *C. jejuni* and an ESBL-producing *E. coli* were evaluated by CFU and v-qPCR. The results showed that freezing and storage in cool conditions are potentially underestimated by CFU but not by v-qPCR. The effect of treatment with peroxyacetic acid on survival was consistently detected by CFU and v-qPCR. v-qPCR analysis detected bacterial survival upon the application of lactic acid, which awaits further analysis. Interestingly, both bacteria showed similar kinetics of inactivation upon the application of reduction strategies, suggesting that *E. coli* might be a complementary hygiene indicator. We conclude that v-qPCR can improve food safety under the consideration of some limitations.

## 1. Introduction

Along the poultry production chain, Thermotolerant *Campylobacter* spp. are frequent fecal contaminants with serious implications for human health. With 220,682 confirmed human cases in 2019, *Campylobacter* infections remain the most common bacterial gastro-intestinal disease in the European Union [1]. *C. jejuni* represents the major reported species. Poultry meat products play a pivotal role as transmitters of sporadic campylobacteriosis cases in the EU [1]. *Escherichia coli* is often used as a hygiene indicator for fecal contamination [2,3]. Furthermore, several *E. coli* strains pose, in addition to *Campylobacter*, a risk for consumers due to their virulence and multi-drug resistance traits, complicating antimicrobial treatments not only of gastric disorders [4,5,6,7].

The viability of bacteria is commonly quantified by their capacity to produce colony forming units (CFU) on agar plates. However, under unfavorable conditions, i.e., upon stress exposure, bacteria may transiently cease cell division, while potentially maintaining infectivity [8,9,10,11,12]. In consequence, CFU may lead to an overestimation of the reduction efficiency of applied decontamination strategies.

Real-time PCR (qPCR) offers a fast, specific, and culture-independent method for bacterial quantification. When applied in combination with a DNA-intercalating agent prior to qPCR, viable/dead differentiation can be achieved in a viability qPCR (v-qPCR). The agent propidium monoazide (PMA) was shown to be passively excluded from viable bacteria while efficiently entering dead cells with compromised membranes [13,14,15]. In dead bacteria, PMA is cross-linked to DNA upon light exposure and subsequently prevents DNA amplification during qPCR.

PCR protocols exist for the detection of *E. coli* without or with the objective to differentiate between viable and dead bacteria. They often utilize the *uidA* sequence [16,17,18,19,20], encoding the enzyme β-glucuronidase, which is commonly used for phenotypic species differentiation. Also genes for serogroups such as O157 [14,21,22,23] are applied. However, for an optimal reduction in the dead cell signal, a suitable amplicon length is necessary in order to maintain efficient amplification and to optimize the chance of PMA binding within the target sequence [24].

In 2018, the European Commission set a process hygiene criterion into force, limiting the contamination level of Thermotolerant *Campylobacter* spp. on chicken carcasses at slaughterhouses [25]. The ultimate goal is to enhance biosecurity at the farm level and to significantly reduce fecal contamination by improving slaughter techniques. However, various decontamination strategies have been discussed, aiming to reduce pathogens in poultry abattoirs [26,27,28,29,30,31,32,33,34,35,36]. For beef, decontamination with 2–5% lactic acid (LA) is legally applied and might complement other hygiene measures [37]. Furthermore, an EFSA scientific opinion presented the option for the decontamination of chicken carcasses by 0.2% peroxyacetic acid (PAA) [38]. Besides chemical decontamination, freezing was also proposed as a reduction strategy for Thermotolerant *Campylobacter* spp. on chicken meat [39].

In this study, we addressed two central aspects. First, we wanted to evaluate diagnostic methods for their suitability to detect viable *Campylobacter* spp. This is most important after stress exposure, e.g., upon chemical decontamination. Second, we wondered whether *E. coli* behaves similar to Thermotolerant *Campylobacter* spp. concerning survival on chicken skin, indicating the “easy-to-handle” *E. coli* as a suitable hygiene indicator. For this purpose, we designed and characterized a novel v-qPCR that is specific for *E. coli* with sufficient viable/dead differentiation power. We also addressed the observation of some authors who discussed false-positive *uidA* signals that were attributed to residual *E. coli* DNA in recombinant Taq polymerases or other reagents [20,40,41]. The novel v-qPCR was applied in parallel with a previously validated v-qPCR for Thermotolerant *Campylobacter* spp. [42] and CFU determination. They were utilized in order to evaluate the survival of both Gram-negative bacterial fecal pathogens after the application of reduction strategies on chicken skin.

## 2. Results

A v-qPCR for the quantification of viable Thermotolerant *Campylobacter* spp. was recently validated [42]. To our knowledge, such a publicly available method, which includes a suitable target length and an internal sample process control (ISPC) is still missing for *E. coli*. Hence, we designed a novel v-qPCR for *E. coli* using a fragment of the *uidA* gene as target. Both methods were then applied to quantify survival in decontamination experiments using chicken skin.

### 2.1. Design of a uidA-Targeting qPCR for Quantifying E. coli

For the reliable detection of *E. coli,* we chose a fragment of the *uidA* gene, encoding the β-glucuronidase, which is commonly used for the phenotypic identification of *E. coli*. We performed a nucleotide BLAST analysis using the *uidA* gene of *E. coli* CFT073 as a template (https://blast.ncbi.nlm.nih.gov/Blast.cgi; last accessed on 1 May 2022). The resulting homologous sequences of each *E. coli* taxon in NCBI and of *Shigella* spp. (in total 372 sequences) were used to generate a consensus sequence of *uidA* (Figure 1). Primer and probe annealing sites were chosen to align within the highly conserved gene segments and were verified towards the *uidA* sequence of our Sanger-sequenced laboratory strains, the reference strain *E. coli* DSM 1103, and the field strain ESBL *E. coli* 10714. The chosen forward primer (*uidA*-F1: 5′-TTATTGCCGGGAAAAGTGTAC-3′) and reverse primer (*uidA*-R2: 5′-AGCCAGTAAAGTAGAACGGTTTG-3′) in combination with the probe (*uidA*-P: 5′-JOE-CTGTTCGCCCTTCACTGCCACTGAC-BBQ-3′) resulted in the amplification of a 513 bp fragment (Figure 1).

### 2.2. Performance of the uidA-Targeting qPCR and Absence of False Positive Signals

For quantification, we produced DNA standards from *E. coli* and tested the novel qPCR for performance. The *E. coli* standards DSM 1103 and ESBL 10714 were amplified with an average efficiency of 95.5 ± 5.7% (n = 21) and 94.7 ± 6.8% (n = 23), respectively (Figure 2A). The IPC-ntb2 [43] was included as an internal amplification control (Figure 2B).

False positive amplification signals may occur due to residual genomic DNA in the IPC-ntb2 plasmid preparation or the heterologous Taq polymerase. They might be an obstacle for the reliable performance of a *uidA*-targeted qPCR, as addressed previously [20,40,41]. However, no false positive signal of *uidA* was detected in the negative controls when 25 copies (or 1000 copies) of the amplification control were applied, indicating the absence of relevant contamination of genomic DNA in the plasmid preparation from the *E. coli* K12 host (Appendix A).

In addition, we compared a native Taq polymerase and several recombinant enzymes or master mixes in terms of efficiently amplifying the *E. coli* DNA standards and the absence of false-positive signals. The native Taq polymerase amplified both *E. coli* genome standards less efficiently. A reduced sensitivity towards lower *E. coli* copy numbers was observed (Appendix A). Besides, the *uidA* channel showed no false-positive amplifications in the negative control samples using either recombinant enzymes or master mixes (Platinum Taq Polymerase, AmpliTaq Gold^®^ DNA Polymerase in TaqMan Gene Expression Master Mix, and HotStarTaq DNA Polymerase in Quanti Tect Multiplex PCR Master Mix), even at elevated concentrations (Appendix A). In further experiments we, therefore, utilized the Platinum Taq Polymerase.

### 2.3. The Novel uidA-Targeted qPCR Is Sensitive and Specific for E. coli/Shigella spp.

The *uidA*-based qPCR for *E. coli* was evaluated for its sensitivity towards the *uidA*-positive target strains. In total, we used 150 field strains (142 *E. coli*) and 19 reference strains (Table 1).

*E. coli* field strains were selected by serotype diversity, virulence gene composition, and biochemical characteristics. The field strains were derived from different food and animal sources (Appendix A), in particular from poultry and cattle but also pigs and wild animals. The tested field strains included 54 enteropathogenic *E. coli* (EPEC) and 46 Shigatoxin-producing *E. coli* (STEC) strains (19 *stx1* only, 21 *stx2* only, and 6 *stx1* and *stx2*). In total, 18 *E. coli* STEC field strains belonged to serotype O157:[H7].

From all 142 *E. coli* field strains and the 4 *E. coli* and 1 *Shigella sonnei* reference strains, positive amplification results were obtained. The novel *uidA*-based qPCR detected all tested *E. coli/Shigella* strains (100% sensitivity), which also included *E. coli* with a reduced or absent phenotypic ß-glucuronidase activity, cleaving MUG.

All tested strains outside the species *E. coli/Shigella* showed negative results (100% specificity). It also accounted for the closely related species *E. albertii*, *E. fergusonii*, and *E. hermannii*, which did not amplify. Hence, the *uidA* target, the designed oligos, and the probe appeared to be suitable for identification of *E. coli* and *Shigella* spp. strains.

### 2.4. Combining the Novel qPCR with a Pre-Treatment Step Using PMA Allows the Differentiation of Viable and Dead E. coli

Viable *E. coli* (DSM 1103 and ESBL 10714) cells were produced by culturing bacteria in BHI until the stationary phase. *E. coli* were killed by 5% H_2_O_2_ under low osmotic conditions at 50 °C. The absence of CFU was checked after enrichment of at least 10^6^ killed bacteria in BHI and subsequent streak-plating. When exposing artificially inactivated *E. coli* to 50 µM PMA, a reduction power of >3.5 log_10_/mL due to the PMA treatment was observed relative to viable bacteria, which is well-suited to differentiate viable from dead cells by v-qPCR (Figure 3).

### 2.5. Storage of Chicken Skin in Refrigeration and Freezing Conditions

Since chicken products are either stored in refrigeration conditions or frozen, we evaluated the cooling/freezing effect on viable *C. jejuni* and *E. coli* over time. For this purpose, we spiked *Campylobacter*-free chicken skin samples with both species and analyzed intact and potentially infectious units (IPIU) by v-qPCR and CFU over a period of 3 days to 3 weeks.

In skin samples stored at 4 °C, initial CFU of *C. jejuni* were maintained stable within 3 days and declined afterwards gradually by around 2 log_10_/mL to reach 3.0 ± 0.7 log_10_/mL after three weeks (Figure 4A, dashed blue line). *E. coli* CFU were stable during 7 days and declined thereafter by around 1 log_10_/mL to reach 3.8 ± 0.3 log_10_/mL after three weeks (Figure 4B, dashed blue line). In contrast to CFU, viable v-qPCR counts (IPIU) from cooled samples remained stable at approximately 5 log_10_/mL for both species, without any decline throughout the experiment (Figure 4, solid blue lines).

Freezing reduced the CFU of *C. jejuni* and *E. coli* after 3 days by 1.5 and 1.2 log_10_/mL, respectively (Figure 4, dashed orange lines). Subsequently, the CFU decreased only marginally. After 3 weeks of frozen storage, the CFU had declined by 2.1 and 1.6 log_10_/mL for *C. jejuni* and *E. coli*, respectively. The loss of IPIU was lower than of CFU. Freezing for three weeks reduced intact *C. jejuni* (IPIU) to some extent, by around 1 log_10_/mL. *E. coli* IPIU were reduced even less, by 0.4 log_10_/mL (Figure 4, solid orange lines). The overall DNA content remained at a constantly high level of around 5 log_10_/mL, indicating the presence of some dead cells (Figure 4, dotted orange lines).

### 2.6. Effect of Chemical Treatment on the Survival of C. jejuni and ESBL-Producing E. coli on Chicken Skin

We quantified the effect of chemical decontamination on chicken skin that was initially spiked in parallel with *C. jejuni* and an ESBL-producing *E. coli* strain (Figure 5).

Skins were treated with the respective chemical solution for 3 min and short-term neutralized in PBS. The final rinse in PBS was analyzed for CFU and by v-qPCR. The chemical solutions PAA (0.5%) and LA (1% and 5%) were evaluated in comparison to a control in which the chemical solution was replaced by water. The initial pathogenic load was assessed by an analysis of untreated samples. The treatment of chicken skin with water for 3 min resulted in a slight but not significant decrease in the initial contamination by ≤0.5 log_10_/mL CFU or IPIU in *E. coli* and *C. jejuni* (Figure 6).

In both species, PAA (0.5%) caused a highly significant complete loss of CFU. Consistently, the absence of a v-qPCR signal or a low residual signal was observed, with reductions of >3.5 log_10_/mL upon PMA treatment. When compared with the exclusively dead cell internal sample process control (ISPC) included in each sample, this indicated the presence of a rest signal from dead cells (Figure 6).

LA significantly reduced CFU of both species compared to water-treated samples in a concentration-dependent manner. Immediate reductions of 1.1 and 1.7 log_10_/mL CFU were observed by treatment with 1% LA for *C. jejuni* and *E. coli,* respectively (Figure 6). When 5% LA was applied, no colonies or only single colonies were detected (≥4 log_10_/mL reduction). In contrast, v-qPCR signals only slightly declined upon LA treatment, with a maximum decrease of 0.7 log_10_/mL for *E. coli* by treatment with 5% LA (Figure 6).

During survival experiments with chemical treatments, the pH was measured of the decontamination solution (right before use), the neutralization bath (after use), and the chicken skin rinse (Table 2). After treatment with 1% LA (and 0.5% PAA), samples passed the neutralization bath, which remained close to neutral in pH (7.0). Consistently, the respective chicken rinses were only slightly acidified, indicating that neutralization showed a sufficient effect. Only in the case of 5% LA treatment, the neutralization bath measured an acidified pH of 4.5–5.5 after use and the actual skin rinses were measured at pH 4.0 to 5.

### 2.7. Long-Term Effects of Lactic Acid Treatment

Since we observed a discrepancy between the CFU and v-qPCR signals upon LA decontamination, we wondered how decontamination affected the CFU and the v-qPCR signals in a long-term experiment. Would CFU recover and how would the signal kinetics of the v-qPCR behave over time, i.e., was a decay of the membrane and a subsequent entry of PMA over time detectable? Therefore, we stored LA-treated samples at 4 °C for 1 day up to 3 weeks and regularly quantified CFU and the v-qPCR signal (Figure 7).

Water-treated control samples (Figure 7, in blue) resembled in pattern in the untreated cooled samples mentioned above (Figure 4). As observed before, the LA treatment led to an immediate rapid loss of CFU, but the v-qPCR signals remained initially high. A recovery of CFU was not observed within the experimental time of three weeks. Instead, we even observed a complete loss of CFU, i.e., 1% LA fully abolished CFU in *C. jejuni* within a day and in *E. coli* within three days.

The viable v-qPCR signals were reduced over time to some extent but inversely to the applied concentration of LA. The decrease in the viable v-qPCR signal was lower after treatment with 5% LA than when 1% LA was applied. In 1% LA-treated samples, the viable v-qPCR signal for *E. coli* was lost after 3 weeks, when it measured 2.3 log_10_/mL for *C. jejuni*. The overall DNA content (without PMA) remained stable for both species and was independent of the intervention and time point tested. These results suggested that after lactic acid treatment, membrane permeability does not indicate bacterial death, which awaits further studies.

## 3. Discussion

Our study was focused on two main aspects. On the one hand, we wondered whether CFU underestimates the survival of Thermotolerant *Campylobacter* spp. on chicken skin upon storage or chemical decontamination. On the other hand, the question was addressed as to whether *E. coli* can serve as a hygiene indicator to mimic Thermotolerant *Campylobacter* on chicken skin. The latter bacterium is much easier to diagnose for routine laboratories and might compensate for technical pitfalls in *Campylobacter* detection.

### 3.1. Establishing a Novel uidA-Based v-qPCR for the Quantification of Viable E. coli

In order to reliably quantify viable *E. coli*, we decided to design a novel *uidA*-based v-qPCR with a target fragment optimized in length. The activity of the targeted enzyme, β-glucuronidase, is also utilized in *E. coli* identification [44,45,46]. Yet, about 3% to 6% of *E. coli* strains remain phenotypically β-glucuronidase-negative [45,47], especially among *E. coli* O157 [48]. Among our tested *E. coli* field strains, cultural β-glucuronidase activity was absent in 22 strains and reduced in 12 strains when evaluated with the substrate 4-methylum-belliferyl beta-D-glucuronide (MUG) by routine testing on Fluorocult *E. coli* 0157:H7 agar. As expected, the *uidA*-based qPCR was superior since all *E. coli* strains tested positive. In an exclusivity and inclusivity study, we ascertained the appropriate sensitivity and specificity of the qPCR.

As usually performed in our *Campylobacter*-qPCR studies, we quantified using a stable DNA standard with a known chromosomal copy number [42,49,50]. This allowed absolute quantification and yielded approximately 95% amplification efficiency (Figure 2). In addition, the v-qPCR was controlled by an ISPC. The ISPC monitors a sufficient reduction in the dead cell signal. It is influenced simultaneously by matrix and potential DNA losses in each individual sample. In this study, the ISPC also guaranteed reliable crosslinking to PMA in samples with slightly decreased pH. With this method at hand, viable and dead *E. coli* were differentiated to a high degree since the signal of dead cells was reduced by at least 3.5 log_10_/mL upon PMA treatment (Figure 3). In combination with our previously developed v-qPCR for Thermotolerant *Campylobacter* spp. [42], we quantified the survival of both *C. jejuni* and an ESBL-producing *E. coli* field strain on chicken skin upon the application of different reduction strategies. Theoretically, the sensitivity of the novel qPCR is 100 copies per ml of chicken skin rinse since our standard curve reliably detected 10 copies per PCR reaction and 10% of the DNA was used per sample.

### 3.2. Effect of Cooling and Freezing

Reductions of 1–2 log_10_ CFU *C. jejuni* on chicken meat have been observed due to freezing. Therefore, freezing is suggested as a suitable reduction strategy [39]. In our study, CFU initially declined in a similar way and maintained over time at a comparable level, as observed previously [29,31,51,52,53].

Freezing and cooling were investigated by culture-independent quantification methods in the past [54]. Moreover, differences in the recovery of *E. coli* on selective and non-selective media were observed after the cooling of meat [55]. The results are in line with our study, with only slight CFU reductions for *C. jejuni* but stable numbers of intact cells detected by v-qPCR under cooling conditions. Moreover, freezing appeared to underestimate the survival of both pathogens, however, to a lower degree than cooling. Our data showed that *C. jejuni* and *E. coli* reacted similarly in physical storage conditions. Hence, cooling and freezing appeared to overestimate reduction strategies. In conclusion, the data suggested an advantage to control the efficiency of those strategies by culture-independent v-qPCR in addition to CFU.

### 3.3. PAA Treatment

The immediate effects of 0.5% PAA and 1% and 5% LA were evaluated in comparison to water-treated and untreated control samples by both CFU and v-qPCR. Intriguingly, the reduction effects were remarkably similar for both Gram-negative species. Water-treated versus untreated samples showed only minor insignificant reductions in both IPIU and CFU. Upon treatment with PAA, IPIU corresponded very well to CFU. PAA is a reagent with oxidizing properties [56], likely leading to loss of cell membrane integrity. It confirms a direct effect on cell membrane integrity and indicates the ultimate loss of cell viability. We conclude that v-qPCR can measure bacterial survival very well upon PAA treatment and can be applied for such samples in practice, which would overcome problems in cultivation and speed up diagnostics. Zhang et al. [57] and Nagel et al. [58] achieved 1–1.3 log_10_ CFU reductions compared to water-treatment on chicken parts with a short-term (~20 s) immersion in 0.12% PAA. As expected, our reductions by 0.5% PAA and 3 min of incubation were much higher, demonstrating cell membrane effects at maximal CFU-reducing conditions. However, these concentrations are not thought to be applied on the product itself, since a maximum of 0.2% PAA is recommended by EFSA for chicken products [38]. Moreover, color changes were observed on chicken skin. However, as a surface disinfectant in poultry production facilities, PAA at high concentrations might be an efficient agent for pathogen reduction. Reductions in *E. coli* CFU were high (>4.2 log_10_) upon the application of 0.5% PAA for 1 min in a surface disinfecting experiment on steel disks without food matrix components [59]. Lower but effective reductions of 1.2–1.3 log_10_ were found in that study in the presence of a matrix-mimicking yeast-albumin agar. Interestingly, our data showed an even higher reduction effect on both *E. coli* and *C. jejuni* by 0.5% PAA in the presence of natural matrix, as found in abattoirs. Our prolonged incubation time may partly explain the differences.

### 3.4. LA Treatment

CFU reductions due to LA treatment observed in previous studies are generally in line with our results. When naturally contaminated broiler carcasses were submerged in 1.5% LA during slaughter, a reduction of 1.2 log_10_ CFU of *C. jejuni* per carcass was measured [60] compared to a water control. Dipping breast pieces with skin for 10 min into 1% or 3% LA reduced *C. jejuni* by 1.3 and 2.0 log_10_ most probable number (MPN), respectively [61]. Reductions on broiler legs were lower. Anang et al. [28] studied *E.*
*coli* O157:H7 re-inoculated on purchased, ethanol/heat-decontaminated chicken breast pieces. CFU were reduced with 2% LA by 1.6 or 2.4 log_10_/g, when treated for 10 or 30 min, respectively.

In our study, LA treatment reduced CFU either partially by 1.1 to 1.7 log_10_/mL (1% LA) or nearly fully (5% LA) after 3 min of dip treatment. We observed that IPIU, in contrast to CFU, remained only marginally affected in both species, upon immediate measurement. These data suggested that LA affected colony formation, but not cell membrane integrity. Theoretically, it leaves the possibility of a part of the cells entering a viable but non-culturable (VBNC) state. Hence, we wondered if, upon storage, sub-lethal injury entailed long-term effects, measurable as accelerated loss of integrity of the membrane.

Indeed, initial CFU loss increased with storage time after 1% LA treatment, until complete loss of culturability. Riedel et al. [62] applied 2.5% LA for 1 min to *C.*
*jejuni* on chicken skin. Compared to water, this dipping treatment initially only slightly reduced *C. jejuni* by 0.7 log_10_ CFU/ml. However, upon additional storage for 24 h at 5 °C, CFU declined by 2.8 log_10_/mL significantly. Thus, our study also confirms a certain accelerated loss of viable cells over time after treatment with 1% LA. This is, as loss of integrity of the inner cell membrane necessarily implies death. However, in 5%-LA-treated chicken skins, not completely neutralised by PBS, loss of intact bacteria was lower over time.

LA is a strong acid with a relatively low pKa value [63]. It is reported to acidify the cytoplasm of neutrophilic bacteria, leading to the denaturation of cell components and the impairment of their functions [64,65]. Indeed, contact to 3% LA led to an immediate drop in internal pH and a progressive loss of CFU in *C. jejuni* [66]. Besides, it was suggested that LA caused permeabilization of the outer membrane of Gram-negative bacteria [67]. Therefore, LA may not have necessarily harmed the inner membrane integrity at first instance. Meanwhile, cell activity was probably strongly abolished. The recovery of the pH gradient after lactic acid exposure and incubation in nutrient broth under microaerobic conditions at 4 °C indicated the integrity of the inner membrane and the principal capacity to regain growth properties in *C. jejuni* [66]. The resuscitation of sublethally injured *E. coli* after exposure to low LA concentrations (<0.1%) was dependent on the medium composition and metal ion availability due to an enhanced oxidative stress response [68,69]. Thus, the selective agar needed for the detection of food pathogens might be suboptimal for the recovery of sublethally injured bacteria. Hence, these aspects call for clarifying the viable state after lactic acid decontamination monitored by DNA-intercalating agents alone. Alternative viability markers and, in particular, animal models may help to answer the question of whether v-qPCR had reached its limit.

## 4. Materials and Methods

### 4.1. Strains and Growth Conditions

*E. coli* strains DSM 1103 (DSMZ strain collection, Braunschweig, Germany) and an extended-spectrum ß-lactamase (ESBL) field strain, specifically ESBL 10714, from chicken meat were grown from a −80 °C cryoculture (MAST Group Ltd, Bootle, UK) in brain–heart infusion (BHI, Becton Dickinson, Franklin Lakes, NJ, USA) at 37 °C and 180 rpm. The preculture was subcultured by an inoculation of an initial OD_600_ of 0.01 and incubation until a stationary phase to an OD_600_ of 4.5 ± 0.5. Bacteria were diluted using phosphate-buffered saline (PBS, pH = 7.4) to an OD_600_ of 1, corresponding to about 9 log_10_ bacteria per ml, and kept at 5 ± 1 °C in a cooling rack (CoolBox^TM^, Corning Inc., Corning, NY, USA) until further use.

Further reference and field strains used for inclusivity and exclusivity evaluations of the qPCR were grown either on tryptic soy agar (TSA, Merck KGaA, Darmstadt, Germany) or Columbia agar supplemented with 5% sheep blood (ColbA, Oxoid, Thermo Fisher Scientific Inc., Waltham, MA, USA). Reference strains, usually obtained as lyophilisates from strain collections, were passaged once and cryo-conserved. Our laboratories are accredited and regularly prove the identity of strains.

*C. jejuni* DSM 4688 (DSMZ strain collection, Braunschweig, Germany) was grown from a −80 °C cryoculture on ColbA at 42 °C in a microaerobic incubator (5% O_2_, 10% CO_2_, 85% N_2_; Binder GmbH, Tuttlingen, Germany) for 24 ± 4 h. The cells derived from an 18 ± 2 h subculture and were suspended in peptone water (PW, Merck KGaA, Darmstadt, Germany) to an OD_600_ of 0.2, corresponding to about 9 log_10_ bacteria per mL [13], and kept at 5 ± 1 °C in a cooling rack until further use.

CFU determination was performed according to ISO 16649-3:2015 for *E. coli* and ISO 10272-2:2017 for *C. jejuni* with the following modifications. Tryptone bile x-glucuronide agar (TBX, Oxoid, Thermo Fisher Scientific Inc., Waltham, MA, USA) was used for spread-plating *E. coli*, which were incubated for 24 h at 37 °C under aerobic conditions. mCCDA was used for *C. jejuni* and incubated for 48–72 h at 42 °C in a microaerobic atmosphere (5% O_2_, 10% CO_2_, 85% N_2_). The MUG activity of *E. coli* strains was checked on Fluorocult *E. coli* O157:H7 Agar (Merck KGaA, Darmstadt, Germany).

### 4.2. DNA Extraction and qPCR

DNA from the survival experiments was extracted using the GeneJet Genomic DNA Purification kit (Thermo Fisher Scientific Inc., Waltham, MA, USA) according to the manufacturer’s instructions and eluted in 100 µL of elution buffer. DNA for the sensitivity and specificity analysis of the *uidA*-based qPCR was extracted from the strains using either the GeneJet Genomic DNA Purification kit or the RTP Bacteria DNA Mini Kit (Stratec, Birkenfeld, Germany). The latter DNA was generally diluted 1:1000 in H_2_O before use. DNA from the samples of the survival experiments was analyzed undiluted for maximum sensitivity.

The qPCR analysis was performed on an ABI Prism 7500 (Life Technologies, Thermo Fisher Scientific Inc., Waltham, MA, USA) in duplicate for each sample. For each reaction, 10 µL of DNA eluate and 15 µL of master mix were used. The three targets, the 16S rRNA of Thermotolerant *Campylobacter*, the 16S rRNA of the ISPC (*C. sputorum*), and the *uidA* of *E. coli,* were each analyzed as duplex qPCRs, including the amplification control IPC-ntb2 [43]. The former two *Campylobacter* targets were performed as previously described [49].

For *E. coli*, the master mix contained 1× Platinum Taq buffer, 2.5 mM MgCl_2_, each dNTP at 0.2 mM, 0.06× ROX, the *uidA*-F1 and *uidA*-R2 each at 500 nM, oligos IPC-ntb2-fw and IPC-ntb2-re at 300 nM, the dark-quenched probes FAM-*uidA*-probe-BBQ and TAMRA-IPC-ntb2-probe-BBQ each at 100 nM (TIB Molbiol, Berlin, Germany), and 2 U of Platinum Taq DNA Polymerase (Invitrogen, Thermo Fisher Scientific Inc., USA). The IPC-ntb2 plasmid was added at 25 copies per reaction to the master mix. An initial denaturation and activation of the hot start polymerase activity at 95 °C for 3 min was followed by 45 cycles of denaturation at 95 °C for 30 s, annealing/elongation and detection of the fluorescent signals at 60 °C for 1 min, and an additional incubation at 72 °C for 30 s. If indicated, other Taq polymerases, such as a native Taq Polymerase from *Thermus aquaticus* (Invitrogen, Thermo Fisher Scientific Inc.), the AmpliTaq Gold^®^ DNA Polymerase (TaqMan Gene Expression Master Mix, Life Technologies, Thermo Fisher Scientific Inc.), and HotStarTaq DNA Polymerase (Quanti Tect Multiplex PCR Master Mix, Qiagen, Hilden, Germany) were used according to the manufacturer’s protocol.

### 4.3. Preparation of Quantification Standards for qPCR

We prepared DNA quantification standards for the *uidA*-based qPCR from the *E. coli* strains DSM 1103 and *E. coli* ESBL 10,714 as previously described for *C. jejuni* [42]. In short, strains from liquid BHI cultures were pelleted by centrifugation for 5 min at 16,000× *g*. The DNA was extracted using the GeneJet Genomic DNA Purification kit according to the manufacturer’s manual. The DNA was quantified using the Qubit 3.0 Fluorometer (Life Technologies, Thermo Fisher Scientific Inc., USA). A volume of 40 µL, containing 26.3 ng of genomic DNA or 2,500,000 *E. coli* chromosomal equivalents per µL, was added to each DNA stable tube (Biomatrica, San Diego, CA, USA). After a thorough mixing of the DNA-stabilizing reagent, 2 µL aliquots were dried overnight in 0.5 mL reaction tubes under the working bench and stored at room temperature. Just before qPCR analysis, an aliquot of *E. coli* DNA was reconstituted in 200 µL of H_2_O (corresponding to 50,000 genome copies per µL). Using 10 ng/µL salmon sperm DNA (Invitrogen, Thermo Fisher Scientific Inc., USA), the reconstituted aliquot was decimally diluted to 50 copies per reaction. A final 5-fold dilution yielded 10 copies per reaction. The five different standards were applied in duplicate during each qPCR run.

For quantifying the genome equivalents of Thermotolerant *Campylobacter* and the ISPC, we applied similar genome standards of the strains *C. jejuni* NCTC 11,168 and *C. sputorum* DSM 5363 as described previously [49]. These quantification standards were applied as 50,000, 5000, 500, 50, and 5 copies per reaction in duplicate. The *uidA* gene is present as a single copy gene in the *E. coli* chromosome, while the 16S rRNA targets are present in the *C. jejuni* and *C. sputorum* genome as three identical copies.

### 4.4. Survival Experiments with Chicken Skin

*Campylobacter*-free skin was produced from ROSS 308 chickens reared and slaughtered at the institute’s facilities. The absence of *Campylobacter* was checked by ISO 10272-1:2017 and by real-time PCR after enrichment in Bolton broth [70]. *E. coli* background flora was quantified on TBX agar with a mean of 3.76 ± 0.46 log_10_ per g of skin. Samples stored at −80 °C were thawed, rinsed twice with MilliQ water, cut to 1 ± 0.1 g portions, and stored at −20 °C until use. For each data point of the experiment (per condition and time point), a separate skin sample of 1 g was used. The CFU and v-qPCR were analyzed from the same skin sample rinse. All skins were retrieved from the same batch of slaughtered chickens. For inoculation, aliquots were placed into 9 × 12.5 cm closable plastic bags (RUBIN, Rossmann, Hannover, Germany) and kept on ice until inoculation.

For survival evaluation, 0.5 mL of each bacterial inoculation suspension of either *E. coli* or *C. jejuni* (7–7.3 log_10_ per ml PW) (see Section 4.1) were both applied to each 1 g skin sample. After manual massaging for one minute, the inoculated skin samples were incubated at room temperature for 30 min.

Chemical solutions were prepared just before application. The pH was measured using pH indicator strips (Merck KGaA, Darmstadt, Germany). The final solutions were as follows: 0.5% peroxyacetic acid (PAA, Wofasteril SC super 1 + 1, Kesla Pharma Wolfen GmbH, Bitterfeld-Wolfen, Germany; pH 5.0) and 1% and 5% lactic acid (LA, Sigma-Aldrich, Merck KGaA, Darmstadt, Germany, pH 2 and pH 1.5, respectively). As a water control, either Milli-Q water (Merck KGaA, Darmstadt, Germany) or tap water was used.

Each skin sample was placed into a separate 20 mL chemical solution for 3 min with an occasional inversion of the tube. Subsequently, the sample was neutralized for 10 s in 20 mL of PBS (single-use). Thereafter, 5 mL of PBS was added to the sample in a new plastic bag, which was massaged for 1 min. Rinse aliquots were taken for CFU and v-qPCR analysis.

The effects of cooling and freezing were evaluated at 5 °C and at −25 °C. The long-term effect of the LA treatment was evaluated by storing the samples at 5 °C for 1 day to 3 weeks.

### 4.5. Viable/Dead Differentiation by v-qPCR

For the quantification of *C. jejuni,* the v-qPCR was conducted as previously described [42]. In short, two 1 mL rinse aliquots were analyzed. One aliquot contained no PMA, which served for the quantification of the overall DNA content. The other aliquot contained PMA for quantifying viable bacteria (IPIU). The ISPC—*C. sputorum* dead cell control—allowed the monitoring of the dead cell signal reduction by PMA as well as DNA losses during DNA extraction for each sample individually. The aliquot measuring the overall DNA content received 10 µL of the *C. sputorum* dead cell control (ISPC_low_) and was centrifuged for 5 min at 16,000× g and 4 °C. After discarding the supernatant, the bacterial pellet was stored at −20 °C until DNA extraction and qPCR analysis (see Section 4.2). The aliquot measuring viable bacteria (IPIU) first received 10 µL of the concentrated ISPC_high_ and was pre-incubated for 10 min at 30 °C with gentle shaking at 700 rpm. Subsequently, 50 µM PMA was added, and the sample was incubated for another 15 min at 30 °C in the dark. Crosslinking occurred afterwards for 15 min using the PhAST Blue photo-activation system at 100% light intensity (GenIUL, Terrassa, Spain). Subsequent to the inactivation of the PMA-reactive azide groups by light, 10 µL of the ISPC_low_ were added, and the sample was centrifuged for 5 min at 16,000× *g* and 4 °C. The bacterial pellets were stored at −20 °C. Control samples accompanied the procedure as described previously [42].

Dead *E. coli* were obtained as follows: A bacterial suspension of 9 log_10_ cells per mL (see Section 4.1) was centrifuged at 16,000× *g* for 5 min. The cell pellet was resuspended in 1:10 diluted PBS supplemented with 5% H_2_O_2_ at a cell density with an OD_600_ of 0.5 and incubated at 50 °C for 60 min. Killed bacteria were centrifuged at 16,000× *g* for 5 min, washed once, and resuspended in PBS to an OD_600_ of 0.1, corresponding to around 10^8^ dead bacteria/mL. The absence of growth was checked by adding 10 µL of this suspension to 10 mL of BHI and incubating overnight at 37 °C with shaking at 180 rpm. The enrichment broth was subsequently streaked on TSA.

### 4.6. Data Handling

The qPCR data were analyzed using a pre-version of the Microsoft Excel data analysis sheet (Stingl et al. 2021) [42]. The CFU and v-qPCR outcomes were log_10_-transformed. The absence of a CFU or qPCR signal was defined as −0.1 log_10_/mL. Statistical significance was tested by one-way ANOVA followed by Tukey’s multiple comparison post hoc test. For each quantification method, intervention groups were analyzed for the statistical significance of their difference compared to the water-treated control group. Statistics were performed in Graph Pad Prism v5.01.

## 5. Conclusions

The current quantification of microbial contaminants in food is limited, as pathogens are underestimated when transiently inactive. Here, we designed and established an ISPC-controlled v-qPCR for *E. coli* and applied a previously validated respective method for *Campylobacter* spp. Using two selected strains, we showed that, depending on the inactivation method used, artificially killed cells can also be principally quantified by v-qPCR. Furthermore, *E. coli* and *C. jejuni* behaved similarly upon immediate quantification. The results of our study are restricted to the tested strains. Therefore, further characterization of multiple *E. coli* and *C*. spp. strains might be considered in future approaches. The method showed high viability upon lactic acid treatment and awaits further studies on the infectivity of these bacteria in animal models or by alternative viability markers. However, we conclude that for improved food safety, v-qPCR methods should complement microbial cultivation.

## Figures and Tables

**Figure 1 pathogens-11-00706-f001:**
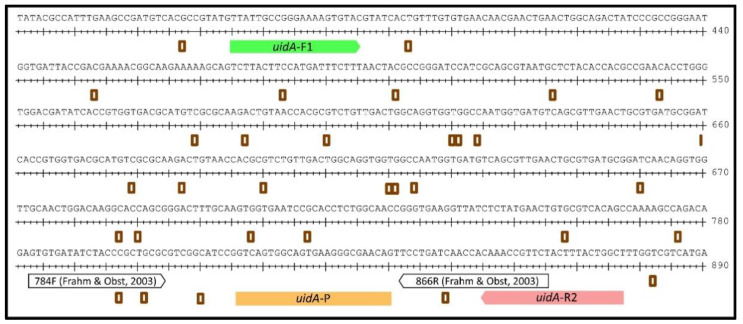
Consensus sequence of the *uidA* fragment (residue 331 to 890) after alignment of 372 *E. coli* and *Shigella* spp. sequences. The novel qPCR targets a 513 bp fragment using the oligos *uidA*-F1 and *uidA*-R2 and the probe *uidA*-P. Example oligo annealing sites of a published qPCR within the depicted region using an 83 bp fragment [18] are also illustrated. Brown squares indicate non-conserved positions in *uidA*. Numbers on the right present the position ruler in *uidA*.

**Figure 2 pathogens-11-00706-f002:**
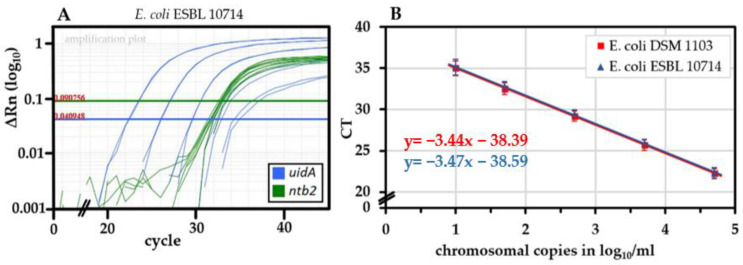
Performance of the *uidA-targeting* qPCR for *E. coli*. (**A**) Example ΔRn amplification plot of the *uidA* target from a decimally diluted genomic DNA standard (*E. coli* ESBL 10714) and the IPC-ntb2 as internal amplification control. (**B**) Correlation between log_10_-transformed Ct values and chromosomal copies of DNA standards (*E. coli* DSM 1103, n = 21 and ESBL 10714, n = 23).

**Figure 3 pathogens-11-00706-f003:**
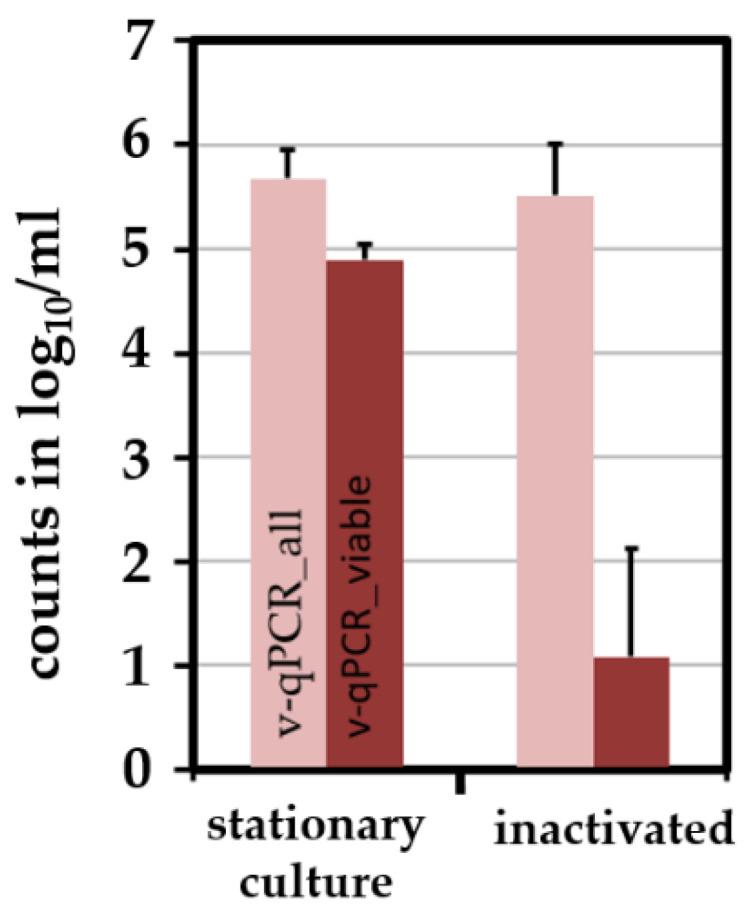
Differentiation of viable and dead *E. coli* adjusted to 10^6^ log_10_/mL PBS, either derived from a stationary culture (n = 4) or after inactivation (n = 15).

**Figure 4 pathogens-11-00706-f004:**
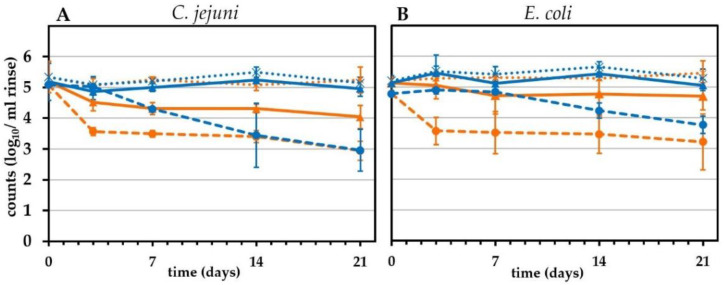
Effect of cooling and freezing on CFU and viability by v-qPCR (IPIU). (**A**) *C. jejuni* DSM 4688; (**B**) *E. coli* ESBL 10714. Orange, −25 °C; blue, 4 °C; dashed lines, CFU; solid lines, viable bacteria (IPIU) counts; dotted lines, total DNA. Means ± standard deviations are illustrated from three independent experiments.

**Figure 5 pathogens-11-00706-f005:**
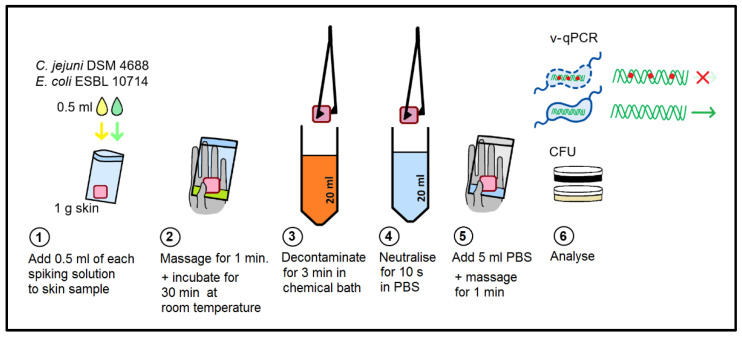
Experimental setup of chemical decontamination of chicken skin.

**Figure 6 pathogens-11-00706-f006:**
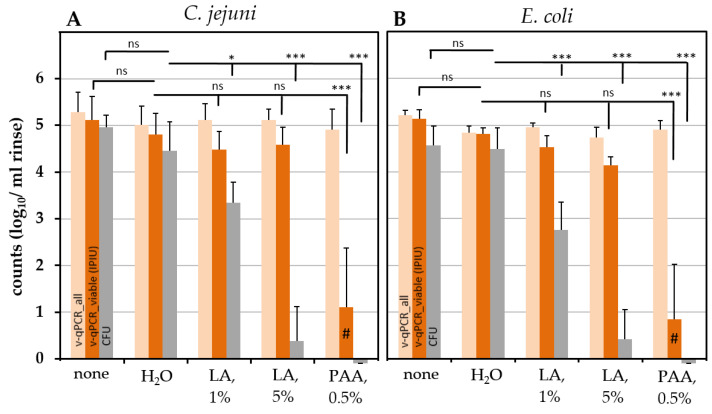
Effect of different chemical treatments of chicken skin on CFU and IPIU analyzed by v-qPCR. (**A**) *C. jejuni* DSM 4688; (**B**) *E. coli* ESBL 10714. Light orange, v-qPCR_all (total DNA); orange, v-qPCR_viable (IPIU); grey, CFU. Significance was evaluated by Tukey’s multiple comparison. Intervention groups were compared to water-treated samples for each analysis method. The means ± standard deviations are depicted from at least three independent experiments. ns, not significant (*p* > 0.05); * 0.05 > *p* > 0.01; *** *p* < 0.001. #, rest signal of dead cells.

**Figure 7 pathogens-11-00706-f007:**
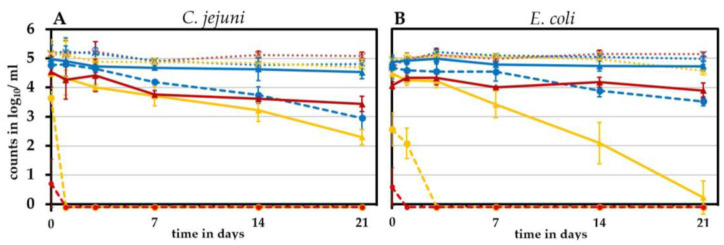
Long-term effects of lactic acid (LA) treatment on cell membrane integrity (v-qPCR) and culturability (CFU) in (**A**) *C. jejuni* DSM 4688 and (**B**) *E. coli* ESBL 10714. Dashed lines, CFU; solid lines, v-qPCR_viable (IPIU); dotted lines, v-qPCR_all (total DNA). Blue, water control; yellow, 1% LA; red, 5% LA. The means ± standard deviations of at least three independent experiments are shown.

**Table 1 pathogens-11-00706-t001:** Results of sensitivity and specificity characterization of the *uidA-based* qPCR.

Reference Strains	Results	Field Strains	Number of Isolates	Results
*Escherichia coli* DSM 1103	positive	*Escherichia coli*, MUG+	108	positive
*Escherichia coli* EDL 933	positive	*Escherichia coli*, MUG(+)	12	positive
*Escherichia coli* DSM 498	positive	*Escherichia coli*, MUG-	22	positive
*Shigella sonnei* DSM 5570	positive	*Enterobacter cloacae* complex	1	negative
*Escherichia albertii* DSM 17582	negative	*Escherichia albertii*	2	negative
*E**scherichia**fergusonii* DSM 13698	negative	*Aeromonas jandaei*	1	negative
*Escherichia hermannii* DSM 4560	negative	*Citrobacter koseri*	1	negative
*Enterococcus faecalis* DSM 20478	negative	*Hafnia alvei*	3	negative
*Klebsiella pneumoniae* DSM 30104	negative			
*Proteus mirabilis* DSM 4479	negative			
*Pseudomonas aeruginosa* DSM 1117	negative			
*Staphylococcus aureus* DSM 1104	negative			
*Salmonella enterica* DSM 11320	negative			
IPC-ntb2 (plasmid)	negative			
*Campylobacter jejuni* DSM 4688	negative			
*Campylobacter jejuni* NCTC 11168	negative			
*Campylobacter sputorum* DSM 5363/ISPC	negative			
*Campylobacter coli* DSM 4689	negative			
*Campylobacter lari* DSM 11375	negative			

MUG, activity to metabolize 4-methylum-belliferyl beta-D-glucuronide; +, positive; −, negative; (+), low activity.

**Table 2 pathogens-11-00706-t002:** pH ^1^ measurements.

Treatment	Decontamination Solution (Before Use)	Neutralization Bath (PBS, after Use)	Chicken Skin Rinse
LA, 1%	2	7	6 (5.5–7)
LA, 5%	1.5	5 (4.5–5.5)	4 (4–5)
PAA, 0.5%	5	7	6.5
MilliQ water	5 (4.5–5)	7.5	7–7.5
tap water	7.5	7.5	7–7.5

^1^ pH was evaluated using pH strips. pH stated as median and range (min–max).

## Data Availability

The datasets generated during and analyzed during the current study can be find in the main text and the Appendix A.

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
