# Peer review of "Chicken Skin Decontamination of Thermotolerant *Campylobacter* spp. and Hygiene Indicator *Escherichia coli* Assessed by Viability Real-Time PCR"

_pathogens, 2022, doi:10.3390/pathogens11060706_

Round 1
Reviewer 1 Report
The paper Chicken skin decontamination of thermophilic Campylobacter spp. and Hygiene Indicator Escherichia coli Assessed by Viability Real Time PCR, propose an interesting study on development of a Viability Real Time Pcr to quantify the viable E. coli after decontamination treatment.
The results of these study are very important for all researchers who works at the validation of the antibacterial product, as Antibacterial peptides, to reduce the contamination of food or food facilities.
In the complex the paper was clear, materials and methods were reliable and robust, Despite I suggest some integration
How many samples were analysed to evaluate the effects of PAA and LA, then the performances of two methods?
Also, I suggest considering the possibility to complete the paper with conclusion to highlight the results of the research.
Author Response
We thank the reviewer for their valuable comments, which we answer point-by-point in the following. Line numbers refer to the document with highlighted track changes.
Reviewer 1
The paper Chicken skin decontamination of thermophilic Campylobacter spp. and Hygiene Indicator Escherichia coli Assessed by Viability Real Time PCR, propose an interesting study on development of a Viability Real Time Pcr to quantify the viable E. coli after decontamination treatment.
The results of these study are very important for all researchers who works at the validation of the antibacterial product, as Antibacterial peptides, to reduce the contamination of food or food facilities.
In the complex the paper was clear, materials and methods were reliable and robust, Despite I suggest some integration
How many samples were analysed to evaluate the effects of PAA and LA, then the performances of two methods?
>>We performed 5 experiments, in which we quantified the immediate effect of chemical decontamination (PAA, LA). In one set of experiment, technical problems with the E. coli detection plates were observed, thus we didn’t include these data for E. coli. Hence, we stated for this figure ‚at least three independent experiments‘. In addition, 3 independent long-term experiments were performed for LA treatment (Fig. 7). For each data point, both methods, CFU and viability qPCR were performed in parallel.
Also, I suggest considering the possibility to complete the paper with conclusion to highlight the results of the research.
>> We thank the reviewer for his notion and added a conclusion part (lines 374-386).
Reviewer 2 Report
The conclusions needs more work. The following sentence "Hence, we conclude that the effect of death could not be readily monitored by DNA inter- calating agents alone, after acid decontamination. Here, v-qPCR seemed to have reached its limit" needs further discussion. What would be the next step on the study of the proposed detection method?
You should reference other similar studies to improve the discussion. Also the limitation of the method are not discussed.
The experiments were designed using a single strain of E coli and Campylobacter. As a standard method, a cocktail of strains is used.
Author Response
Reviewer 2
The conclusions needs more work. The following sentence "Hence, we conclude that the effect of death could not be readily monitored by DNA inter- calating agents alone, after acid decontamination. Here, v-qPCR seemed to have reached its limit" needs further discussion. What would be the next step on the study of the proposed detection method?
You should reference other similar studies to improve the discussion. Also the limitation of the method are not discussed.
>> We added 5 more references (ref No. 63-66 and 68-69, marked in yellow), including studies, which suggest viability and/or resuscitation after lactic acid treatment in C. jejuni and E. coli. These papers were discussed in terms of LA concentration used and the viability marker monitored. Furthermore, we appointed next steps/ future perspectives (lines 361-378). We also re-formulated the sentences, which mentioned the “limitations”, with more focus on future studies to clarify the apparent difference between CFU and IPIU (lines 22-23, 259-260).
The experiments were designed using a single strain of E coli and Campylobacter. As a standard method, a cocktail of strains is used.
>> During performance testing of the designed E. coli v-qPCR a specificity study was performed using more than 140 field and several reference strains. The reviewer is right that the results drawn for survival of C. jejuni and E. coli upon decontamination is limited. The study is more focussed in establishing a dual quantification strategies using v-qPCR and to show advantages and limits of the alternative method. We added a sentence following the notion of the reviewer that “Using two selected strains, we showed that, depending on the inactivation method used, also artificially killed cells can be principally quantified by v-qPCR. Furthermore, E. coli and C. jejuni behaved similarly upon immediate quantification. The results of our study are restricted to the tested strains. Therefore, further characterization of multiple E. coli and C. spp. strains might be considered in future approaches. ” In lines 378-382.
Reviewer 3 Report
Chicken Skin Decontamination of Thermophilic Campylobacter spp. and Hygiene Indicator Escherichia coli Assessed by Vi- 3 ability Real-Time PCR
In this study the authors designed and characterized a novel v-qPCR, specific for E. coli with sufficient viable/dead differentiation power to evaluated diagnostic methods for their suitability to detect viable Campylobacter spp. And studied if E. coli behaves like thermophilic Campylobacter spp. concerning survival on chicken skin.
Overall, the paper is explained well with the detailed of methodology. However, further details of the statistical analysis are needed.
What are the sensitivity and specificity of this procedure and how it may affect your findings?
Please check the structures of sentences
Ex: Lines 68-69/ 258-259
Please explain the implication of these findings.
Author Response
Reviewer 3
Chicken Skin Decontamination of Thermophilic Campylobacter spp. and Hygiene Indicator Escherichia coli Assessed by Vi- 3 ability Real-Time PCR
In this study the authors designed and characterized a novel v-qPCR, specific for E. coli with sufficient viable/dead differentiation power to evaluated diagnostic methods for their suitability to detect viable Campylobacter spp. And studied if E. coli behaves like thermophilic Campylobacter spp. concerning survival on chicken skin.
Overall, the paper is explained well with the detailed of methodology. However, further details of the statistical analysis are needed.
>>We added some further details on the test method in legend of Figure 6. Further information may be found in the methods‘ section, in paragraph „5.6 ‚Data handling‘ “ (former 4.6.).
What are the sensitivity and specificity of this procedure and how it may affect your findings?
>> We added the sentence “Theoretically, the sensitivity of the novel qPCR is 100 copies per ml of chicken skin rinse, since our standard curve reliably detected 10 copies per PCR reaction and 10 % of the DNA was used per sample.” (lines 288-290). The specificity was already addressed in line 150.
Please check the structures of sentences, Ex: Lines 68-69/ 258-259
>> We separated the long phrase in lines 68-69 into two short sentences and shortened the phrase in lines 258-9 (now 260).
Please explain the implication of these findings.
>> A conclusion part was added, in which we also address the implications (lines 374-386).
Reviewer 4 Report
The article has scientific quality, and I am quite sure that the results will be highly interesting for the readers
Nevertheless, some minor considerations.
Please consider changing the title of the article. The title needs to be changed by giving a concise assertion of the study done.
Introduction needs to be modified by focusing more on research goals to show the importance of this study: on why the two specific bacteria were selected and what their relationship is?
- 67: “In this study, we asked two central questions”….. . Maybe we posed two….
L 133: “Shigella spp.”. Shigella spp.
Table 1: Please write the full name of: C. jejuni DSM 4688, C. jejuni NCTC 11168, C. sputorum DSM 5363 / ISPC, C. coli DSM 4689, C. lari DSM 11375, and afterwards use abbreviation and italics, in the rest of the text.
Section Material & Methods: The depiction of the methodological approach is very confusing, which does not help any new researcher to follow it and carry it out. Which, on the other hand, is one of the requirements for any new article. Please the authors consider to revise it.
Questions:
-Where were the used reference strains deposited? Was the first passage from a freeze-dried vial?
- How many skin samples were tested?
- The Campylobacter-free chicken skin samples were spiked with both species at the same time, or separately for each reference strain?
- L 437-439: “Campylobacter-free skin was produced from ROSS 308 chicken reared and slaughtered at the institute’s facilities. Absence of Campylobacter was checked by ISO 10272- 1:2017 and by real-time PCR after enrichment in Bolton broth”. You also checked for the presence of other bacteria?. Were the samples also E. coli free?
L 379-380: “Tryptone bile x-glucuronide agar (TBX, Oxoid, Thermo Fisher Scientific Inc., USA) ..”. What is happening here? Please fix it.
It’s crucial to discuss the methodological limitations of the article. A «limitation» section should be structured.
The section conclusions is missing. Please, complete section “Conclusion” with information on the practical application of the obtained results of this study.
Author Response
The article has scientific quality, and I am quite sure that the results will be highly interesting for the readers
Nevertheless, some minor considerations.
Please consider changing the title of the article. The title needs to be changed by giving a concise assertion of the study done.
>> In the title we address i) both investigated pathogens – thermophilic Campylobacter spp. and Escherichia coli as hygiene indicator for fecal contamination (if Campylobacter spp. is absent in the avian gut, e. g. in winter times, E. coli can serve as hygiene indicator), ii) the matrix used for challenging the novel v-qPCR for E. coli and the already validated one for C. spp. and iii) the idea of alternative analysis of viability by v-qPCR. As we are limited in title length, more specific notions cannot be included.
Introduction needs to be modified by focusing more on research goals to show the importance of this study: on why the two specific bacteria were selected and what their relationship is?
>> In the introduction we have the notion that “Escherichia coli is often used as hygiene indicator for fecal contamination [2,3].” (lines 36-37) And later on that the goal of the study is ” Second, we wondered whether E. coli behaves similar to thermophilic Campylobacter spp. concerning survival on chicken skin, indicating the “easy-to-handle” E. coli as suitable hygiene indicator.” (lines 70ff.) Thus, the relationship is that both are fecal bacteria but that most of the laboratories have the capacity to handle E. coli but not Campylobacter. Thus, we wanted to address the two, which is mentioned in lines 78-79 “They [both species] were utilized in order to evaluate the survival of both Gram-negative bacterial fecal pathogens after application of reduction strategies on chicken skin.”
67: “In this study, we asked two central questions”….. . Maybe we posed two….
>> We changed the phrase into “we addressed two central aspects”.
L 133: “Shigella spp.”. Shigella spp.
>> According to the format rules of the Journal, subtitles have to be completely in italics. Thus, we fear we cannot change it here. All other “Shigella spp.” are in the right format throughout the manuscript text body.
Table 1: Please write the full name of: C. jejuni DSM 4688, C. jejuni NCTC 11168, C. sputorum DSM 5363 / ISPC, C. coli DSM 4689, C. lari DSM 11375, and afterwards use abbreviation and italics, in the rest of the text.
>> We changed the names accordingly, including Escherichia strains for consistency.
Section Material & Methods: The depiction of the methodological approach is very confusing, which does not help any new researcher to follow it and carry it out. Which, on the other hand, is one of the requirements for any new article. Please the authors consider to revise it.
Questions:
-Where were the used reference strains deposited? Was the first passage from a freeze-dried vial?
>> We added to chapter 5.1 the notion: “Reference strains usually obtained as lyophilisates from strain collections were once passaged and cryo-conserved. Our laboratories are accredited and regularly prove the identity of strains.”
- How many skin samples were tested?
>> We inserted a sentence to chapter 5.4: “Per data point of experiment (per condition and time point), a separate skin sample of 1 g was used. CFU and v-qPCR were analysed from the same skin sample rinse. All skins were retrieved from the same batch of slaughtered chicken.”
- The Campylobacter-free chicken skin samples were spiked with both species at the same time, or separately for each reference strain?
>> In order to clarify the inoculum we added “were both applied to each 1 g skin sample” to line 477.
- L 437-439: “Campylobacter-free skin was produced from ROSS 308 chicken reared and slaughtered at the institute’s facilities. Absence of Campylobacter was checked by ISO 10272- 1:2017 and by real-time PCR after enrichment in Bolton broth”. You also checked for the presence of other bacteria?. Were the samples also E. coli free?
>> No, it is possible to produce Campylobacter-free chicken skin but (at least with normal slaughtering techniques) no E. coli free samples. We checked the load of the background E. coli flora on chicken skin and observed around 3-4 log10 per g. Thus, our inoculum exceeded the natural flora multifold and rendered the experiment feasible for our interpretations. We added one sentence “E. coli background flora was quantified on TBX agar with a mean of 3.76 ± 0.46 log10 per g skin.” to lines 468-469.
L 379-380: “Tryptone bile x-glucuronide agar (TBX, Oxoid, Thermo Fisher Scientific Inc., USA) ..”. What is happening here? Please fix it.
>> We do not understand the reviewers comment. The full sentence appears to us correct (TBX is the abbreviation of the full name, Oxoid as part of Thermo Fisher Scientific Inc. (Incorporated), USA): “Tryptone bile x-glucuronide agar (TBX, Oxoid, Thermo Fisher Scientific Inc., USA) was used for spread-plating E. coli, which were incubated for 24 h at 37 °C under aerobic conditions.”
It’s crucial to discuss the methodological limitations of the article. A «limitation» section should be structured.
The section conclusions is missing. Please, complete section “Conclusion” with information on the practical application of the obtained results of this study.
>>We added a separate Conclusion chapter (lines 374-386) and added more discussion on the methods limitations and on the putative sublethal state after lactic acid treatment (see also 363-373).